# SARS-CoV-2 Omicron BA.1 Variant Infection of Human Colon Epithelial Cells

**DOI:** 10.3390/v16040634

**Published:** 2024-04-19

**Authors:** Avan Antia, David M. Alvarado, Qiru Zeng, Luis A. Casorla-Perez, Deanna L. Davis, Naomi M. Sonnek, Matthew A. Ciorba, Siyuan Ding

**Affiliations:** 1Department of Molecular Microbiology, Washington University School of Medicine in St. Louis, St. Louis, MO 63110, USA; avan.antia@wustl.edu (A.A.); qzeng@wustl.edu (Q.Z.); 2Inflammatory Bowel Diseases Center, Division of Gastroenterology, Department of Medicine, Washington University School of Medicine in St. Louis, St. Louis, MO 63110, USA; alvaradodm@wustl.edu (D.M.A.); d.l.davis@wustl.edu (D.L.D.); sonnekn@wustl.edu (N.M.S.)

**Keywords:** SARS-CoV-2 Omicron BA.1, SARS-CoV-2 WA1, SARS-CoV-2 Delta, human primary colonoids, intestinal infection, interferon responses, intestinal permeability, spike processing

## Abstract

The Omicron variant of SARS-CoV-2, characterized by multiple subvariants including BA.1, XBB.1.5, EG.5, and JN.1, became the predominant strain in early 2022. Studies indicate that Omicron replicates less efficiently in lung tissue compared to the ancestral strain. However, the infectivity of Omicron in the gastrointestinal tract is not fully defined, despite the fact that 70% of COVID-19 patients experience digestive disease symptoms. Here, using primary human colonoids, we found that, regardless of individual variability, Omicron infects colon cells similarly or less effectively than the ancestral strain or the Delta variant. The variant induced limited type III interferon expression and showed no significant impact on epithelial integrity. Further experiments revealed inefficient cell-to-cell spread and spike protein cleavage in the Omicron spike protein, possibly contributing to its lower infectious particle levels. The findings highlight the variant-specific replication differences in human colonoids, providing insights into the enteric tropism of Omicron and its relevance to long COVID symptoms.

## 1. Introduction

The Coronavirus Disease 2019 (COVID-19), caused by the Severe Acute Respiratory Syndrome Coronavirus 2 (SARS-CoV-2), has and continues to contribute to illness and death on a global scale [1]. Notably, the emergence of multiple variants of concern (VOCs) with mutations in their spike protein, enhancing their interaction with the human angiotensin converting enzyme 2 receptor (ACE2), has added complexity to the ongoing pandemic. Following the four VOCs—Alpha (B.1.1.7), Beta (B.1.351), Gamma (P.1), and Delta (D.1, B.1.617.2), Omicron B.1.1.529 (comprising subvariants such as BA.1, XBB.1.5, and the recent JN.1) ascended to predominance in early 2022, instigating renewed public apprehension.

The Omicron variant, characterized by an unprecedented 50 mutations, including over 30 within the spike protein compared to the ancestral strain, exhibits distinctive genomic features contributing to its attenuated replication in the lung parenchyma relative to the ancestral SARS-CoV-2 and other variants [2]. Notably, several of these mutations have been associated with compromised neutralization of Omicron variants by sera from both pre-Omicron convalescent individuals and vaccinated individuals, resulting in a reduction in the efficacy of antibodies utilized in clinical settings [3,4,5,6]. Additionally, Omicron demonstrates an increased affinity for the ACE2 receptor [7]. Intriguingly, there is evidence suggesting that Omicron may preferentially utilize the endosomal entry pathway over the TMPRSS2-mediated plasma membrane fusion pathway, thereby prompting further inquiries into Omicron’s tropism and infectivity [8,9]. These multifaceted molecular characteristics of the Omicron variant underscore the need for comprehensive investigations to elucidate its distinct behavior and its implications for the dynamics of the COVID-19 pandemic.

COVID-19 induces dysregulation of the gastrointestinal tract (GI), leading to abdominal symptoms. Up to 70% of COVID-19 patients manifest at least one GI symptom, such as nausea, vomiting, and diarrhea. Furthermore, viral RNA has been detected in stool specimens for up to 7 months post-clearance of the virus from the lungs [10]. Thus, it is crucial to understand the virulence and host immune responses of Omicron in various target organs, including the GI tract. Given the pivotal role of ACE2 in facilitating cellular entry for SARS-CoV-2, various cell types expressing ACE2 are deemed susceptible to viral infection and serve as potential target cells. Consequently, variations in ACE2 expression levels may influence the route of viral invasion and the pathogenicity of the virus. Previous research, including our own, has demonstrated elevated expression levels of ACE2 and TMPRSS2/4, the host receptor and proteases essential for SARS-CoV-2 cellular entry, respectively, in the human small and large intestines [11,12,13]. Notably, the intestine, rather than the lung, has been identified as the organ with the highest expression levels of the viral receptor ACE2, underscoring the significance of investigating intestinal involvement in SARS-CoV-2 infection [14,15]. This finding correlates with the natural enteric pathogenicity observed in several animal coronaviruses (CoVs), which are known to cause gastrointestinal (GI) diseases and spread via the fecal–oral route [16]. Furthermore, considerable quantities of SARS-CoV-2 RNA have been detected in stool specimens from COVID-19 patients, although the isolation of infectious virus particles from feces has shown variability across studies [17,18]. Nevertheless, the presence of SARS-CoV-2 RNA in fecal samples raises concerns regarding the potential for fecal–oral transmission of the virus.

Additionally, GI disorders may persist beyond the acute phase of infection, contributing to the post-acute sequelae of COVID-19, known as long COVID, which includes symptoms such as fatigue, post-exertional malaise, memory impairment, and other neurocognitive deficits [19,20,21]. Notably, studies have reported a reduction in serotonin production, primarily synthesized in the intestine, following SARS-CoV-2 infection, resulting in memory impairment in a murine model [21]. Intriguingly, analyses of published metabolomics datasets from various COVID-19 patient cohorts have consistently demonstrated a marked depletion in serotonin levels among the metabolites identified, which reinforces the role of the GI tract in SARS-CoV-2 infection [22].

In this study, we show that in the context of donor intrinsic genetic heterogeneity, the SARS-CoV-2 Omicron variant infects human colonoids similarly, if not less effectively, than the ancestral WT (WA1) strain or the Delta variant. This investigation establishes a foundational understanding for subsequent studies aimed at elucidating the mechanisms governing intestinal infection and pathogenesis by the Omicron variant.

## 2. Materials and Methods

All study procedures and reagents were approved by the Washington University IRB (#202011003). Primary colon epithelial cells (colonoids) were derived from healthy donor biopsies and cultured as previously described [23]. Each SARS-CoV-2 isolate and passage was confirmed by RNA sequencing (Appendix A). Supernatant from infected transwell colonoid monolayers was titrated by focus forming assays. Fixed monolayers were stained for SARS-CoV-2 nucleocapsid (N), actin, and DAPI prior to confocal imaging. Expression levels of SARS-CoV-2 N, GAPDH (Glyceraldehyde 3-phosphate dehydrogenase), interferon lambda (IFNL3), interferon beta (IFNB), and MX1 were quantified by RT-qPCR (primers and probes in Appendix A). HEK293-hACE2-TMPRSS2 cells were transfected with plasmids encoding variant spike proteins and plasmids encoding GFP and imaged at 24 h post-transfection for syncytia formation. HEK293-hACE2 cells were transfected with plasmids encoding variant spike proteins and plasmids encoding empty vector control or V5-tagged host proteases TMPRSS2 or furin and analyzed for spike cleavage using Western blot at 24 h post-transfection.

*Colonoid culture, infection, and harvesting*: Primary intestinal epithelial cells were derived from the colon biopsies of four patients (211A, 235A, 251A, 262A) (Appendix A). Briefly, each biopsy was minced with scissors before digestion with dispase. The tissue was strained through a 70 µm filter and cells were embedded in Matrigel (3D culture, Sigma, Cat: CLS354234-1EA, St. Louis, MO, USA) and maintained in 50% L-WRN conditioned medium supplemented with 10 µM each Y-27632 (R&D Systems, Cat: 1254/10 mg, Minneapolis, MN, USA) and SB431542 (R&D Systems, Cat: 1614/10 mg), as described previously [24]. For 2D cultures, transwell (Corning, Cat: 3470, Corning, NY, USA) devices with polyester membranes with 0.4 µm pore size were pre-treated with 1:10 Matrigel in PBS for 30 min at 37 °C. Colonoids were seeded into transwells and maintained for 7 days in 50% L-WRN conditioned medium containing 10 μM Y-27632. Differentiation was performed using Dulbecco’s modified Eagle medium/F12 (Sigma, Cat: D6429-500ML) supplemented with 20% FBS (Sigma, Cat: F6178), L-glutamine, penicillin/streptomycin, and 10 μM Y-27632 for 3 days before infection. Infections were conducted apically at an MOI of 0.01 for 1 h at 37 °C, after which the viral inoculum was removed, replaced with differentiation media, and incubated for 24 h. For viral RT-qPCR quantification, cell lysates were harvested in TRIzol and RNA was extracted according to the manufacturer’s protocol.

*Viral propagation and sequencing*: All virus passages were conducted in Vero E6 TMPRSS2 cells, as previously described [25]. Viral stock was harvested in TRIzol and RNA was extracted according to the manufacturer’s protocol. SARS-CoV-2 sequences were enriched using the ARTIC v4.1 primer set for SARS-CoV-2 viral enrichment and sequenced on the Illumina NovaSeq platform. Output sequences were trimmed and aligned with the SARS-CoV-2 reference (NC_045512.2) by The Genome Technology Access Center at Washington University in St. Louis.

*Immunofluorescence*: Transwells were fixed in 4% paraformaldehyde (PFA) for 20 min at room temperature and stained with anti-SARS-CoV-2 N antibody (40588-T62; Sino Biological, Beijing, China), phalloidin (Alexa Fluor 647), and DAPI for immunofluorescence confocal imaging.

*Focus Forming Assay*: Vero E6-TMPRSS2 cells were seeded at 2.5 × 10^4^ cells/well in 96-well plates and grown overnight in Dulbecco’s Modified Eagle Medium (Thermo Fisher, Waltham, MA USA) supplemented to contain 10% heat-inactivated fetal bovine serum, 10 mM HEPES, and 100 U/mL penicillin/100 and U/mL streptomycin to reach confluency. Cells were transferred to a biosafety level 3 (BSL-3) facility for infection with viral supernatant collected from the apical compartment of infected transwells. Cells were incubated with 100 μL of viral supernatant at 37 °C for 1 h, after which viral supernatant was removed, replaced with 100 μL of a prewarmed overlay mixture of 2X MEM + 4% FBS with 2% methylcellulose in a 1:1 ratio, and incubated for 30 h at 37 °C [25]. After overlay removal, cells were washed 6 times with PBS and fixed with 4% PFA for 20 min, before removal of PFA and replacement with PBS. Plates were removed from the BSL-3 facility and permeabilized with PBS + 0.1% Triton-X100 for 10 min at room temperature, washed twice with PBS + 0.1% Tween-20 (PBST), and blocked with PBST with 1% BSA and 10% FBS for 1 h at room temperature. Upon removal of the blocking buffer, cells were incubated overnight at 4 °C with a primary anti-SARS-CoV-2 N (40588-T62; Sino Biological) antibody (1:1000 dilution in PBST + 1% BSA). Cells were later washed in PBST, incubated for 45 min at room temperature with secondary Goat anti-Rabbit IgG (Heavy chain) Superclonal Recombinant Secondary Antibody, HRP (Thermo Fisher Scientific A27036) (1:1000 dilution in PBST + 1% BSA), washed with PBST, and developed with AEC substrate kit, Peroxidase (HRP) and methods (Vector Laboratories SK-4200, Newark, CA, USA). Foci were quantified under an ECHO Revolve microscope (Discover Echo, San Diego, CA, USA).

*Syncytia assay*: HEK293-hACE2-TMPRSS2 cells were transfected using Lipofectamine 3000 reagent and manufacturer’s methods (Thermo Fisher L3000015) with plasmids encoding variant spike proteins (WT pTwist-SARS-CoV-2 Δ18, plasmid #164436; pTwist-SARS-CoV-2 Δ18 B.1.617.2v1, plasmid #179905; pTwist-SARS-CoV-2 Δ18 B.1.1.529, plasmid #179907, all acquired from Addgene, Watertown, MA, USA) and plasmids encoding EGFP-N1 and imaged at 24 h post-transfection for syncytia formation.

*Spike cleavage assay*: Plasmids encoding variant spike proteins (WT pTwist-SARS-CoV-2 Δ18, plasmid #164436; pTwist-SARS-CoV-2 Δ18 B.1.617.2v1, plasmid #179905; pTwist-SARS-CoV-2 Δ18 B.1.1.529, plasmid #179907) and plasmids encoding either EGFP-N1 (control), pcDNA3.1/nV5-TMPRSS2 [12], or pLenti6.3/V5-furin host proteases (generated in-house via Gateway cloning) were co-transfected into HEK293-hACE2 cells. Lipofectamine 3000 reagent and manufacturer’s methods (Thermo Fisher L3000015) were used for all transfections. At 24 h post-transfection, cells were washed with PBS, lysed with RIPA buffer (Thermo Fisher Scientific 89901) supplemented with Halt Protease Inhibitor Cocktail (100X) (Thermo Fisher Scientific 78429), and incubated on ice for 10 min. Cell lysates were then subjected to centrifugation at 13,500 RPM for 10 min at 4 °C to remove cell debris and nucleus. Protein samples were boiled in 2X Laemmli Sample Buffer (Bio-Rad, San Francisco, CA, USA) containing 5% β-mercaptoethanol at 95 °C for 5 min. Prepared samples were run in 4–12% Mini-PROTEAN TGX Precast protein gels (Bio-Rad 4561085) and transferred onto nitrocellulose membranes using the Bio-Rad wet/tank blotting system. Membranes were blocked in 5% BSA in TBS + 0.1% Tween-20 (TBST) at room temperature before incubation with primary antibodies at 4 °C overnight. Membranes were then washed three times with TBST and incubated in secondary antibodies diluted in 5% BSA in TBST at room temperature for 1 h. Finally, membranes were washed with TBST and visualized by using ChemiDoc imaging system (Bio-Rad).

*Primary antibodies*: SARS-CoV-2 Spike S2 Rabbit pAb (Sino Biological 40590-T62), SARS-CoV-2 Spike S1 RBD Rabbit pAb (Sino Biological 40592-T62), V5-Tag Rabbit mAb (Cell Signaling Technology 13202S, Danvers, MA, USA), and GAPDH (BioLegend 631402, San Diego, CA , USA). *Secondary antibodies*: Goat anti-Rabbit IgG (Heavy chain), Superclonal Recombinant Secondary Antibody, HRP (Thermo Fisher Scientific A27036).

*Statistical Analysis*: All data were subjected to the Shapiro-Wilk test for normality and were subjected to parametric or non-parametric analysis of variance (ANOVA) as appropriate. Statistics were performed using GraphPad Prism 10.

## 3. Results

### 3.1. Omicron Variant SARS-CoV-2 Infects Human Colon-Derived Organoids

Healthy donor-derived colonoids were seeded onto 2D transwell monolayers, differentiated, and apically infected by the SARS-CoV-2 WA1 strain to determine the optimal time points for intracellular viral RNA measurement. We found that for colonoids obtained from two individual donors (211A and 251A), the levels of viral RNA increased by 1–2 logs within the first 24 h and represented the majority of viral replication (Appendix A). In the next set of experiments, we extended our analysis to colonoids derived from four individual donors and different SARS-CoV-2 strains including WT, Delta, or Omicron BA.1 using an MOI of 0.01 and a 24 h infection period. Compared to WT and Delta, Omicron showed significantly increased replication as measured by intracellular viral RNA levels in 211A and 251A (Figure 1). Despite inter-individual differences in infectivity with each variant, a consistent trend was observed in the colonoids derived from donor 262A (Figure 1). To corroborate active replication, we conducted immunofluorescence to visualize intracellular SARS-CoV-2 N antigens in infected colonoids (Figure 2). We additionally performed a focus-forming assay to measure the amount of infectious SARS-CoV-2 progenies secreted into the apical colonoid supernatants. Intriguingly, Omicron produced comparable (211A and 251A) or numerically lower levels of infectious viruses than Delta and WT (235A and 262A) (Appendix A). This higher ratio of viral RNA to infectious virus titer suggests that Omicron is potentially less infectious in the intestine in some individuals than WT and Delta.

### 3.2. Interferon Responses in Human Colonoids to SARS-CoV-2 Variants

Type I and III interferons (IFNs) are induced in response to various pathogens, playing an important role in initiating the expression of interferon-stimulated genes (ISGs) crucial for antiviral immune responses at the mucosal barrier. In the context of COVID-19, a correlation has been identified between its severity and deficiencies in type I IFN, while stronger type I IFN responses are associated with asymptomatic infection [26]. Notably, treatment with type III interferon has been reported to accelerate the clearance of SARS-CoV-2 [27]. To investigate the type I and type III responses in human colonoids, we challenged them with multiple SARS-CoV-2 variants and subsequently quantified the RNA expression levels. We found that Omicron induced variable, but statistically similar, levels of type III IFN (IFNL3) expression, compared to the other SARS-CoV-2 variants, but not different from the mock infected control (Figure 3A). There was a minimal induction of type I IFN (IFN-β) and MX1, a canonical ISG highly induced by both type I and III IFNs, at 24 h post-infection (Figure 3B,C). This may be attributed to a dampened IFN response at the early stages of SARS-CoV-2 infection, as evidenced by detectable IFN-β and MX1 expression at 48 and 72 h post-infection (Appendix A).

### 3.3. SARS-CoV-2 Infection Does Not Compromise the Integrity of Primary Human Intestinal Colonoids

In addition to assessing innate immune responses, we investigated the potential impact of multiple VOCs of SARS-CoV-2 infection on the integrity of the intestinal barrier, as it represents a plausible mechanism for the occurrence of diarrhea. To address this, we measured transepithelial electrical resistance (TEER) in primary human enteroids multiple times post-infection. Notably, our findings reveal that throughout the course of infection, none of the tested SARS-CoV-2 strains resulted in a reduction in TEER relative to the baseline, indicating the absence of epithelial barrier compromise (Figure 4). This suggests that factors other than a leaky gut may underlie the development of diarrheal symptoms during SARS-CoV-2 infection.

### 3.4. The Omicron Variant Exhibits Lower Syncytia Formation and Impaired Spike Processing in Human Colonoids

To understand the molecular basis and reconcile the disparity between the high abundance of viral RNA and the low virus titer observed in the Omicron variant within human intestinal epithelial cells, we conducted an investigation into the nature of the Omicron spike protein. Utilizing HEK293 cells engineered to stably express human ACE2 and TMPRSS2, we ectopically expressed spike proteins from various SARS-CoV-2 variants [12]. Our findings revealed that the Omicron spike protein induced the formation of fewer and smaller syncytia by fluorescence microscopy compared to either WT or Delta variants (Figure 5), consistent with a recent study [8]. Given the prevalence of increased numbers and sizes of syncytia observed with the wild-type (WT) and Delta variants, we hypothesized that the diminished syncytium formation and reduced fusogenicity observed with the Omicron variant could be linked to a lower efficacy of spike protein cleavage. In line with our prior findings, we found that the cleavage levels of S1, by TMPRSS2 and furin proteases, were notably higher for the Delta variant compared to WT. Similarly, the cleavage levels of S2 were also higher for Delta compared to WT. In contrast, we noted an inefficient cleavage of the Omicron spike protein into both the S1 and S2 compared to Delta and WT variants (Figure 6). Our findings indicate that the Omicron spike protein undergoes less efficient cleavage and exhibits reduced fusogenicity compared to the spike proteins of both the Delta variant and early-pandemic SARS-CoV-2. This may suggest possible attenuation of Omicron upon viral egress, when processed into mature infectious viruses. Taken together, these data collectively suggest that although Omicron may have similar infectivity as the WT and Delta variants in the first round of infection, it is less effective at cell-to-cell spread and producing viral progenies.

## 4. Discussion

In this study, we found that the SARS-CoV-2 Omicron BA.1 variant effectively infects healthy donor-derived colonoids, producing high levels of intracellular viral RNA in some donors, but comparably lower levels of infectious particles. We also found that Omicron induced a weak IFN response after 24 h, possibly due to reduced recognition by cytosolic sensors or viral antagonism of IFN responses. Interestingly, at 48 and 72 hpi, there is an increase in the production of IFNB, which may suggest that this pathway could be relevant in restricting SARS-CoV-2 virus infection in primary human intestinal cells. In concordance with our findings, previous studies show that the type 1 interferon pathway show a more potent effect restricting the rotavirus infection in human enteroids than the type III interferon pathway [28]. This suggests that type I IFN may be the critical IFN for limiting enteric virus replication in the human intestine. To date, only one study has compared the infectivity of SARS-CoV-2 variants in human enteroids. Using spike pseudotyped lentiviral viruses and a luciferase-based reporter assay to quantify infection, the authors observed a 2.5- and 5-fold higher infection of colonoids with the Omicron pseudotype spike compared to Delta and D614G spikes, respectively [11]. The use of an authentic SARS-CoV-2 virus and quantitation of both viral RNA and infectious particles in our study may help explain this potential discrepancy. Furthermore, our observations of diminished syncytia formation and impaired spike processing in the Omicron variant support the reduced infectivity observed in human colonoids.

We found that barrier function, as measured by TEER, was not impaired by any of the variants in human-derived intestinal epithelial monolayers. This is in contrast with previous reports indicating alterations in tight junctions [29] and changes in permeability by SARS-CoV-2 [30]. However, these studies were conducted using Vero E6 cells and human pulmonary microvascular endothelial cells. Potential tissue- or segment-specific effects may explain the differences in this observation. Volcic and colleagues made the preliminary observation that the Omicron variant of SARS-CoV-2 caused less damage to mucosal integrity and barrier function in a colon epithelial model based on the Caco-2 cell line [31]. In this study, they also demonstrated that TMPRSS2 inhibition with Camostat effectively prevented SARS-CoV-2 infection in their model. In contrast, they found that cathepsin inhibition, which is relevant to the endosomal SARS-CoV-2 entry mechanism, did not prevent barrier disfunction. These results should be verified in organoid models that more accurately reflect normal colon physiology and diversity among individuals.

GI symptoms in COVID-19 patients are strikingly frequent and have generated great interest for understanding how SARS-CoV-2 interacts with intestinal physiology. Disease states and commonly prescribed anti-inflammatory drugs can modulate intestinal ACE2 and protease expression, which potentially may have altered infectivity and disease severity in the initial waves of the pandemic [23,32,33]. Further, COVID-19 causes gut microbial dysbiosis and microbial diversity does not recover to pre-infection levels, even 6 months post-initial infection [34]. These investigations also revealed that changes in the gut microbial composition of COVID-19 patients are characterized by an enrichment of opportunistic pathogens, such as *Streptococcus* and *Clostridium*, alongside a depletion of beneficial commensals, such as *Bifidobacterium* and *Lactobacillus* [35]. In addition, both viral infection and perturbed gut microbiota have the potential to disturb the normal function of the gut barrier and lead to impaired intestinal permeability and the degradation of tight junction proteins, such as occludin, junctional adhesion molecule-A, and claudin-1. This disruption can promote the translocation of opportunistic microorganisms into the bloodstream, ultimately triggering systemic inflammation [36]. Furthermore, it is posited that such dysbiosis may be one potential contributor to long COVID. It is now of great interest to examine these possibilities in further studies in the context of Omicron and other SARS-CoV-2 variants. Due to the presence of viral RNA in stool and wastewater, there was concern for potential fecal–oral SARS-CoV-2 transmission [17]. Here, we show that at least in the colon, the Omicron variant efficiently replicates but does not produce more infectious virus particles than other variants at 24 hpi. However, we cannot disregard the possibility that virus production by the Omicron variant may increase at later time points post-infection, thereby potentially raising concerns regarding gastrointestinal (GI) virus shedding.

In summary, our study establishes SARS-CoV-2 variant specific replication differences in human colonoids and stresses the importance of genetic heterogeneity and donor variation as a key factor to consider in organoid studies. Our findings of high viral RNA and low infectious virus levels are also reminiscent of the discrepancy seen in COVID-19 patient stool samples, further highlighting enteroid systems as a useful tool to interrogate virus-host interactions. As the pandemic evolves, there is already evidence for future variants that can have unique features of transmission and pathogenesis. As a potential viral reservoir, it is crucial to understand the molecular mechanisms of Omicron infection in the intestines and further examine the relevance to long COVID symptoms.

## Figures and Tables

**Figure 1 viruses-16-00634-f001:**
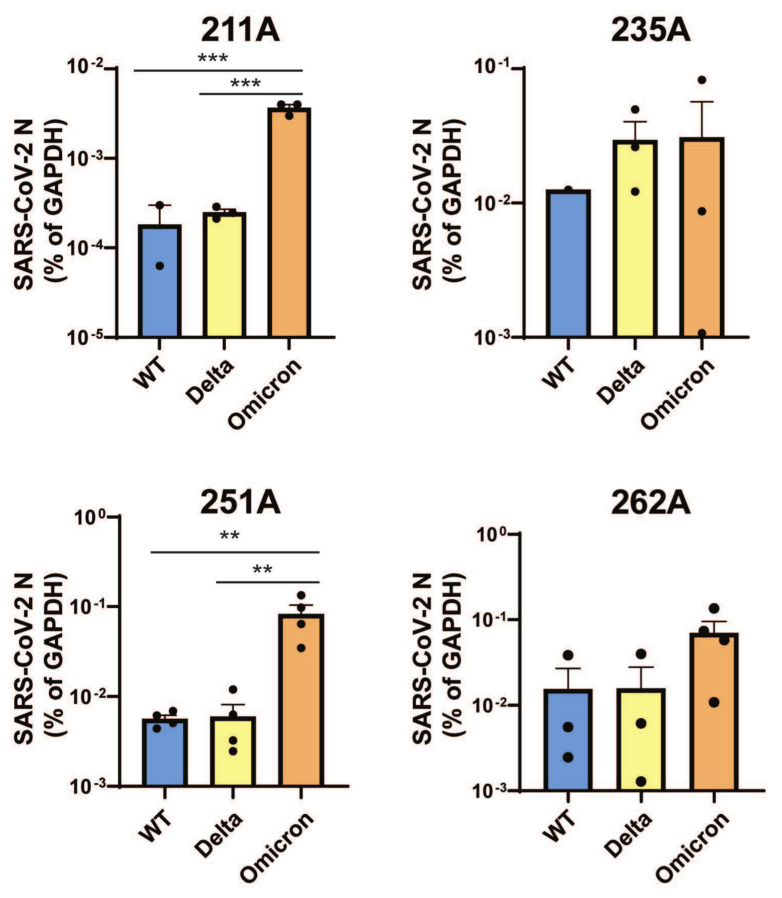
**SARS-CoV-2 RNA increase in donor derived colonoids.** Colonoid lines in 2D transwell monolayers derived from four donors were infected with indicated SARS-CoV-2 variants at an MOI of 0.01. RNA was harvested at 24 h post infection and SARS-CoV-2 N level was quantified by RT-qPCR and normalized to GAPDH. (Mean with SEM (standard error of the mean)), one-way ANOVA with Tukey’s multiple comparisons test. ** *p* < 0.01, *** *p* < 0.001).

**Figure 2 viruses-16-00634-f002:**
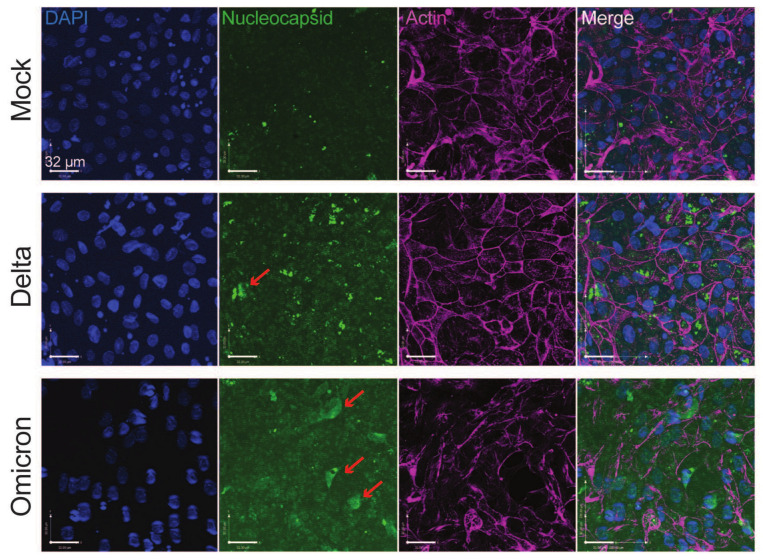
**SARS-CoV-2 viral antigen staining in donor derived colonoids.** Colonoid 262A monolayers were infected by Delta or Omicron at an MOI of 0.01 and fixed at 24 h post infection. Z-stacked confocal microscopic images were acquired and stained for SARS-CoV-2 N (green), actin (violet), and DAPI (blue). Merge image demonstrates intracellular viral nucleocapsid staining. Red arrows indicate intracellular viral staining. Scale bar: 32 μm.

**Figure 3 viruses-16-00634-f003:**
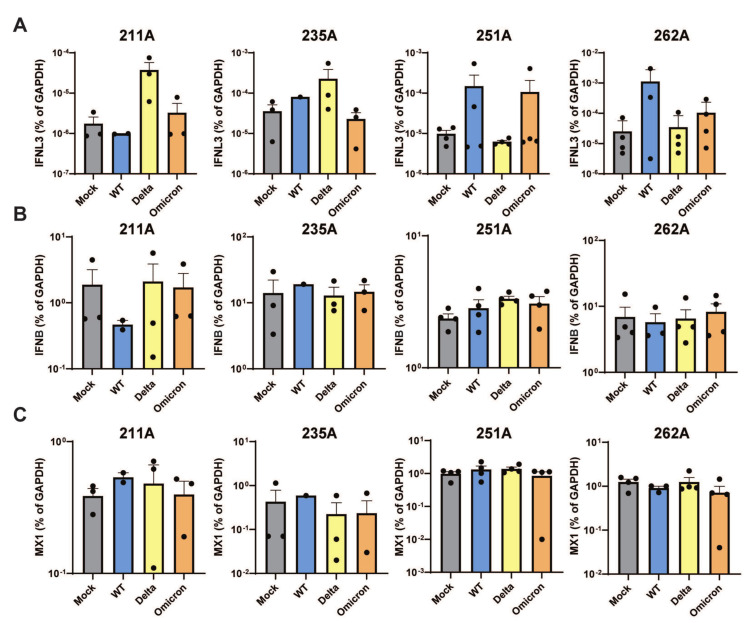
**Host interferon responses to SARS-CoV-2 infection.** Colonoid lines in 2D transwell monolayers derived from four donors were infected with indicated SARS-CoV-2 variants at an MOI of 0.01. RNA was harvested 24 h post infection and levels of IFNL3 (**A**), IFNB (**B**), and MX1 (**C**) were quantified by RT-qPCR and normalized to GAPDH (mean with SEM).

**Figure 4 viruses-16-00634-f004:**
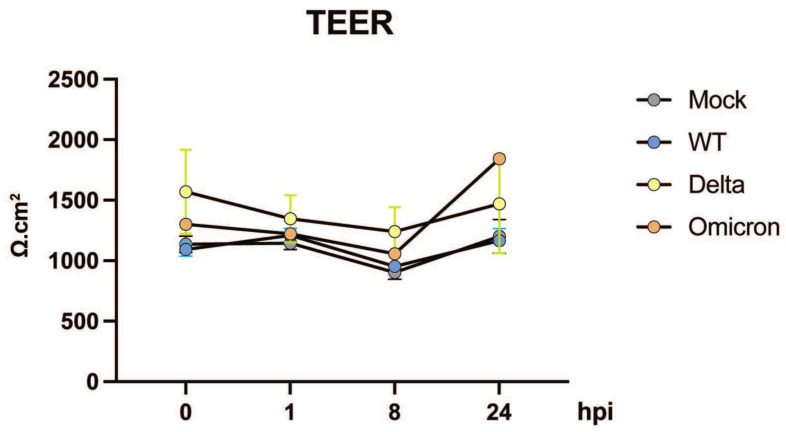
**Intestinal epithelium permeability post SARS-CoV-2 infection.** A human duodenal line in 2D transwell monolayers was infected with indicated SARS-CoV-2 variants. TEER was measured at 0, 1, 8, and 24 h post infection.

**Figure 5 viruses-16-00634-f005:**
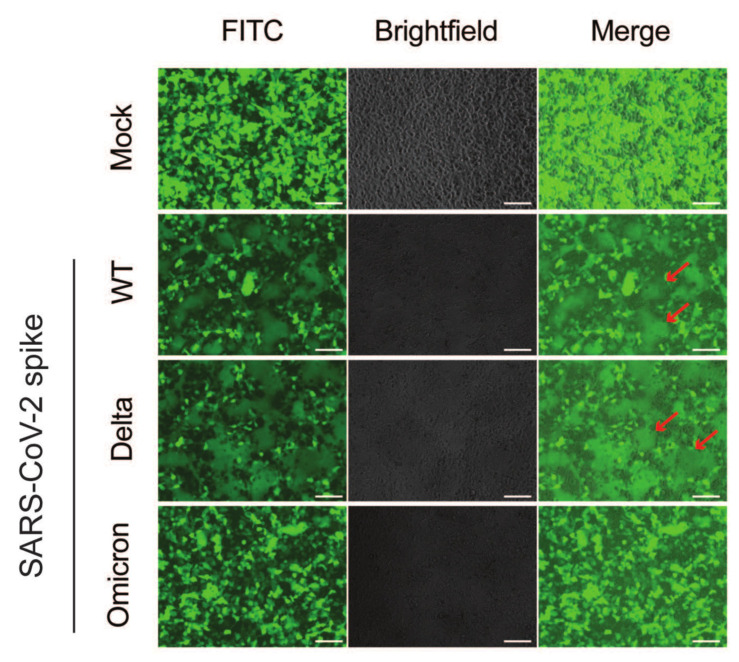
**SARS-CoV-2 variant spike-mediated cell fusion.** HEK293-hACE2-TMPRSS2 cells were co-transfected with plasmids expressing GFP and spikes from the indicated SARS-CoV-2 variants. Microscopic images were captured at 24 h post-transfection. Syncytia are indicated with red arrows. Scale bar: 100 μm.

**Figure 6 viruses-16-00634-f006:**
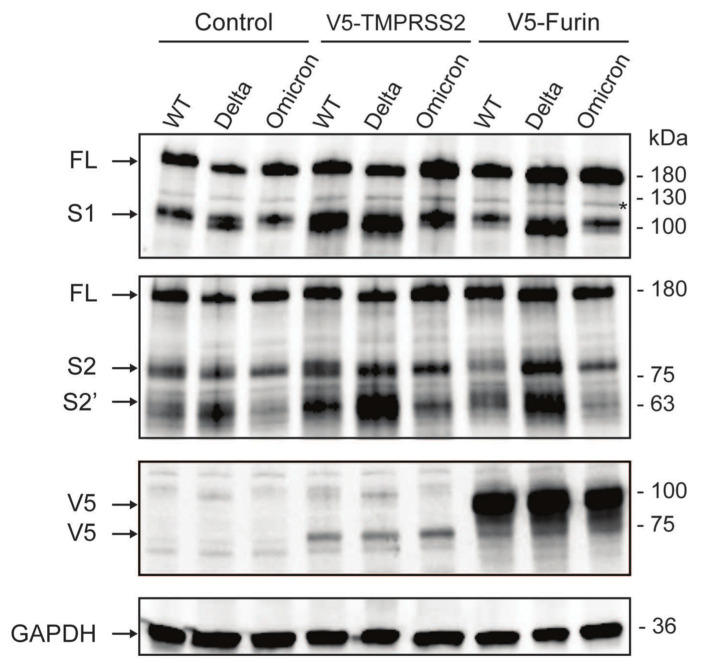
**SARS-CoV-2 variant spike cleavage by host proteases.** HEK293-hACE2 cells were co-transfected with plasmids expressing spikes from the indicated variants and V5-tagged host proteases or empty vector control. Lysates were harvested 24 h post-transfection and analyzed using Western blot. Black arrows indicate full length (FL) uncleaved spike, cleaved S1, S2, and S2′, V5-tagged host proteases (TMPRSS2 and furin), and GAPDH. Asterisk indicates non-specific band.

## Data Availability

All data, analytic methods, and study materials will be made available to other researchers upon request.

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
