# Peer review of "SARS-CoV-2 Omicron BA.1 Variant Infection of Human Colon Epithelial Cells"

_viruses, 2024, doi:10.3390/v16040634_

Round 1
Reviewer 1 Report
Comments and Suggestions for Authors
What subvariant of Omicron did you use?
It would be interesting to know the background of the patients from whom the colonoids were obtained.
Derived from the previous question, perhaps a control of a known cell line is needed and compared with that of the patients since there are disparities in the IFN responses.
In the case of TEER, did you find longer infection times?
Do you know information about SARS-CoV-2 infection in cells from different portions of the intestine?
The authors should discuss the relationship between the different variants used and the intestinal microbiota, an issue that could also affect intestinal integrity and complement their results.
Author Response
Comments and Suggestions for Authors
- What subvariant of Omicron did you use?
- Subvariant BA.1 was used as indicated in the title and is now also added to line 164.
- It would be interesting to know the background of the patients from whom the colonoids were obtained.
- Below please find the information about the background from the colonoid donors.
|
Patient ID |
Patient Diagnosis (N = non-IBD/healthy, CD = Crohn's, UC = ulcerative colitis, FAP = Familial aden. Poly. |
Center Biopsy Collected At |
Intestinal region |
Age @ Access |
Sex 1=Male 2=Female |
Race 1=White 2=Black 3=Asian 4=Hispanic 5=Multiple/Other 6=Native American |
|
|
H211 |
N |
WashU |
Rectum |
55 |
2 |
2 |
|
|
H235 |
N |
WashU |
Rectum |
51 |
2 |
2 |
|
|
H251 |
N |
WashU |
Rectum |
63 |
1 |
1 |
|
|
H262 |
N |
WashU |
Rectum |
56 |
2 |
1 |
- Derived from the previous question, perhaps a control of a known cell line is needed and compared with that of the patients since there are disparities in the IFN responses.
- We appreciate the suggestion about including a regular cell line, however considering the tremendous individual variability in the susceptibility to SARS-CoV-2 replication, no statistical difference in the IFN responses was observed at 24 hpi. This highlights the importance of our study, which is more physiologically relevant due to genetic heterogeneity and distinguishes itself from the established cell line analysis.
- In the case of TEER, did you find longer infection times?
- Unfortunately in our studies we focused the analysis only at 24 hpi.
- Do you know information about SARS-CoV-2 infection in cells from different portions of the intestine?
- We thank for this question and our team has investigated previously the ancestral SARS-CoV-2 replication in human primary intestinal cells derived from ileum and determined a productive infection in mature enterocytes. (Zang R et al., TMPRSS2 and TMPRSS4 promote SARS-CoV-2 infection of human small intestinal enterocytes, Science Immunology, 2020). Comparing the ancestral SARS-CoV-2 replication in ileum and colon, it seems that the virus replication is more efficient in the ileum than in the colon.r
- The authors should discuss the relationship between the different variants used and the intestinal microbiota, an issue that could also affect intestinal integrity and complement their results.
- Per reviewer’s suggestion, we now included a paragraph in the discussion (lines 346-357).
Reviewer 2 Report
Comments and Suggestions for Authors
This paper addresses a problem that has been only seldom investigated; the consequences of the replication of the SARS-Cov-2 in the intestinal tract. The use of infectious viruses for this purpose is certainly more informative than using “pseudoviruses”.
Main remarks:
IFN production. One would expect a production of IFN-III in infected cells as it is the case for Delta in 211A (and some isolated positive results with the other viral strains). Results with IFN-I are inconclusive after 24h which is confirmed by the supplemental figure. However, the production of IFNB is surprising for epithelial cell and should be commented.
How it comes that such differences are observed between the results with virus strains and cell lines as well as inside assays ? What would have been observed on day 2 and 3?
The sentence in the conclusion “Here, we show that at least in the colon, the Omicron variant does not produce more infectious particles, potentially reducing concern for GI virus shedding” is puzzling: It seems that the virus replicates efficiently in the cells thus the production of virions may be delayed as the study was performed over 24h.
The sentence “Due to the presence of viral RNA in stool and wastewater, there was concern for potential fecal-oral SARS-CoV-2 transmission”. For me it may rather be an aerial contamination from stools aerosols (as described for SARS-Cov-1) as the free virus is likely rapidly destroyed in the gastrointestinal tract. Unfortunately this mode of contamination as not seriously been investigated. In infected individual, how the virus have access to the intestinal epithelium (massive ingestion?)
Additional remarks:
Line 186 it is the N protein which accumulate as the presence of nucleocapsids is not demonstrated.
What is S2'?
Indicate for the reagents the purveyor as usual (name, city, country or state for the USA)
Syncytia should have been shown using cell coloration as bright fields are barely informative.
GAPDH and SEM to be explicated
Figure 6: red arrows are missing.
Author Response
This paper addresses a problem that has been only seldom investigated; the consequences of the replication of the SARS-Cov-2 in the intestinal tract. The use of infectious viruses for this purpose is certainly more informative than using “pseudoviruses”.
- We appreciate the reviewer’s positive summary of our work. To address the concerns and weaknesses, we have performed additional experiments and included new experimental data elaborated below.
Main remarks:
- IFN production. One would expect a production of IFN-III in infected cells as it is the case for Delta in 211A (and some isolated positive results with the other viral strains). Results with IFN-I are inconclusive after 24h which is confirmed by the supplemental figure. However, the production of IFNB is surprising for epithelial cell and should be commented.
- We thank the reviewer for this astute observation and we have now included the comments on IFNB in the discussion section (lines 311-317).
- How it comes that such differences are observed between the results with virus strains and cell lines as well as inside assays ? What would have been observed on day 2 and 3?
- One plausible explanation for the difference in interferon response among virus strains, particularly with the Omicron variant, may be due from its extensive array of mutations, which could confer resistance to immune responses. Indeed, reports indicate that Omicron induces lower levels of interferon compared to the Delta and WT variants. Additionally, Omicron surpasses both WT and Delta in its ability to withstand the antiviral effects induced by IFN-alpha (Shalamova L et al., Omicron variant of SARS-CoV-2 exhibits an increased resilience to the antiviral type I interferon response. PNAS Nexus, 2022). We predict that at 48 and 72 hours hpi, Delta will exhibit either higher or comparable levels of type I interferon induction compared to WT, and higher levels compared to Omicron. Similarly, in the type III interferon pathway, we anticipate that Delta will demonstrate the highest levels of induction at 48 and 72 hpi, consistent with observations at 24 hpi. Consequently, we predict that at 48 and 72 hpi, Omicron will exhibit the lowest levels of induction in both the type I and type III interferon pathways.
- The sentence in the conclusion “Here, we show that at least in the colon, the Omicron variant does not produce more infectious particles, potentially reducing concern for GI virus shedding” is puzzling: It seems that the virus replicates efficiently in the cells thus the production of virions may be delayed as the study was performed over 24h.
- We appreciate the suggestion and have changed the conclusion to: “Here, we show that at least in the colon, the Omicron variant efficiently replicates but does not produce more infectious virus particles than other variants at 24 hpi. However, we cannot discard the possibility that virus production by the Omicron variant may increase at later time points post-infection, thereby potentially raising concerns regarding gastrointestinal (GI) virus shedding.
- The sentence “Due to the presence of viral RNA in stool and wastewater, there was concern for potential fecal-oral SARS-CoV-2 transmission”. For me it may rather be an aerial contamination from stools aerosols (as described for SARS-Cov-1) as the free virus is likely rapidly destroyed in the gastrointestinal tract. Unfortunately this mode of contamination as not seriously been investigated. In infected individual, how the virus have access to the intestinal epithelium (massive ingestion?)
- We thank the reviewer for this question. There have been a few reports of the isolation of infectious virus particles in feces. In addition, both oral or nasal inoculation of the virus result in virus replication in the GI tract (Lee AC et al., Oral SARS-CoV-2 Inoculation Establishes Subclinical Respiratory Infection with Virus Shedding in Golden Syrian Hamsters. Cell Rep Med, 2020). Thus, potentially contaminated food with feces from infected people (as occur regularly with enteric viruses) or via aerosols may infect naive individuals.
Additional remarks:
- Line 186 it is the N protein which accumulate as the presence of nucleocapsids is not demonstrated.
R= N stand for nucleocapsid.
- What is S2'?
- It is the cleavage site processed by the transmembrane serine protease 2 (TMPRSS2) on the host cell membrane, leading to the exposure of the fusion loop for initiating membrane fusion.
- Indicate for the reagents the purveyor as usual (name, city, country or state for the USA)
- We have added the proper information as suggested.
- Syncytia should have been shown using cell coloration as bright fields are barely informative.
- We are thankful for this suggestion, although the syncytia were pointed with arrows in the merged micrographs to facilitate easy visualization.
- GAPDH and SEM to be explicated
- Both have now been defined in lines 93 and 205, respectively.
- Figure 6: red arrows are missing.
- We have changed the arrows to the black color.
Round 2
Reviewer 1 Report
Comments and Suggestions for Authors
The authors have covered all my doubts. The article is ready for publication.